# Harmful cultural practices during perinatal period and associated factors among women of childbearing age in Southern Ethiopia: Community based cross-sectional study

**Haimanot Abebe** [ID]*, **Girma Alemayehu Beyene**[☉], **Berhanu Semra Mulat**[☉]

Department of Public Health, College of Health Sciences and Medicine, Wolkite University, Wolkite, Ethiopia

☉ These authors contributed equally to this work.
* haimanotabebe78@gmail.com

**Data Availability Statement:** All relevant data are within the manuscript and its Supporting Information files.

## Abstract

### Introduction

Although the maternal mortality ratio has decreased by 38% in the last decade, 810 women die from preventable causes related to pregnancy and childbirth every day, and two-thirds of maternal deaths occur in Sub-Saharan Africa alone. The lives of women and newborns before, during, and after childbirth can be saved by skilled care. The main factors that prevent women from receiving care during pregnancy and childbirth are harmful cultural practices. The aim of this study was to assess the level of harmful cultural practices during pregnancy, childbirth, and postnatal period, and associated factors among women of childbearing age in Southern Ethiopia.

### Methods

A community-based cross-sectional study design was conducted in the Gurage zone, among representative sample of 422 women of reproductive age who had at least one history of childbirth. A simple random sampling technique was used to recruit participants. Data were collected by six experienced and trained data collectors using a pretested structured questionnaire with face to face interviews. Harmful cultural practices are assessed using 11 questions and those who participate in any one of them are considered as harmful cultural practices. Descriptive statistics were performed and the findings were presented in text and tables. Binary logistic regression was used to assess the association between each independent variable and outcome variable.

### Results

Harmful cultural practices were found to be 71.4% [95%CI, 66.6–76.0]. The mean age of study participants was 27.6 (SD ± 5.4 years). Women with no formal education [AOR 3.79; 95%CI, 1.97–7.28], being a rural resident [AOR 4.41, 95%CI, 2.63–7.39], having had no antenatal care in the last pregnancy [AOR 2.62, 95%CI, 1.54–4.48], and pregnancy being attended by untrained attendants [AOR 2.67, 95%CI, 1.58–4.51] were significantly associated with harmful cultural practice during the perinatal period.

**Funding:** No external funding source is obtained for this study.

**Competing interests:** The authors have declared that no competing interests exist.

**Abbreviations:** ANC, Antenatal Care; MCH, Maternal and Child Health; NGO, Non-Governmental Organization; SPSS, Statistical package for social science.

## Conclusion

In this study we found that low maternal education, rural residence, lack of antenatal care and lack of trained birth attendant were independent risk factors associated with women employing harmful cultural practices during the perinatal period. Thus, strong multi-sectoral collaboration targeted at improving women's educational status and primary health care workers should take up the active role of women's health education on the importance of ANC visits to tackle harmful cultural practices.

## Introduction

Even though the maternal mortality ratio has decreased by 38% in the last decade, daily 810 women died from preventable causes related to pregnancy and childbirth, 94% of them occurred in low and middle-income countries and two-thirds of maternal deaths occurred in Sub-Saharan Africa alone[1]. Ethiopia is one of the countries with the highest maternal mortality; according to the 2016 Ethiopian Demographic and Health Survey report, the pregnancy-related mortality ratio was 412 maternal deaths per 100,000 live births [2]. The lives of women and newborns before, during, and after childbirth can be saved by skilled care. The main factors that prevent women from receiving care during pregnancy and childbirth are cultural beliefs and practices among others [1, 2].

Harmful cultural practices during the perinatal period which include pregnancy, childbirth, and postnatal period refer to deep-rooted traditional practices that adversely affect physical, sexual, psychological well-being, and/ or violate human rights, socio-economic participation, and benefits of women, children, and societies at large. The types and prevalence of these practices vary among regions, cultural settings, religious values, and cultural heritage [3, 4].

In developing countries, cultural beliefs and practices may avert women from accessing antenatal, delivery, and postnatal care. It also has a significant influence on the place of delivery and increases the probability of home delivery [5–10]. For instance, A qualitative study conducted in Jordan revealed that women believe that childbearing is a blessing of Allah, a time for special maternal care, a time for maternal self-renewal, a time for maternal spiritual purification, and a time to prepare for the sacrifices of motherhood. Hence, to gain women's trust in maternity services, nurses need to address mothers' cultural and spiritual needs and meet these needs respectfully [11].

Studies in Asia, Latin America and Africa have shown that a range of restrictive practices through pregnancy and the postpartum period were revealed, and a wide range of good foods and bad foods continued to have currency through the perinatal continuum, with little consensus between groups of what was beneficial versus harmful [12]. As a result, there are varieties of nutritious food items that are avoided during pregnancy and the postnatal period increasing vulnerability of women to malnutrition [13, 14]. A similar, qualitative study conducted in Uttarakhand also showed that a wide variety of cultural practices have been identified during various stages of the perinatal period. Most of the participants (80%) expressed that families believed that pregnant women should not eat green vegetables, yam, pulses, red grams, papaya, and mangoes and that they should eat less during pregnancy [15].

According to the 2016 Ethiopian Demographic and Health Survey, only 32% of women had four or more antenatal care visits during their pregnancy, 73% of pregnant women gave birth

at home and 81% did not receive postnatal checkup. Access to antenatal care, delivery, and postnatal care can be improved by enhancing women's education and empowerment [2].

Harmful cultural practices are common among women and children in Ethiopia. Surveys conducted in Ethiopia indicated that the prevalence of culturally harmful practices during pregnancy ranged from 37–85% [3, 4]. The commonly mentioned harmful cultural practices includes the food restriction and taboos, abdominal and uterine massage, home delivery, avoiding colostrum, cutting the umbilical cord by unsterile sharp materials, delaying initiation of breastfeeding, early bath, giving butter and/or water for newborn, using of "*Koso*" (traditional herb). Maternal age, women's empowerment, educational status of the women, parity of the women, societal awareness, religion, low economic status of women and girls, imbalanced gender relations, and distance from a health facility were among the factors identified to affect the harmful cultural practices [2, 4, 14–21].

To improve maternal health, barriers that limit access to quality maternal health services must be identified and addressed, one of the barriers being harmful cultural practices [1]. In Ethiopia, data on harmful traditional practices during pregnancy, childbirth, and postnatal period are not well understood and no study has been conducted in the study setting to assess the problem. Thus, this study was attempts to determine the level of culturally harmful practices during the perinatal period and the associated factors among reproductive-age women in Southern Ethiopia.

The findings of the study could have potential relevance and significance in understanding the magnitude of the problem, making evidence-based decisions, and taking appropriate actions by developing a health care plan, policy, and programs to resolve the problem.

## Materials and methods

### Study area and period

The study was conducted in the selected woreda of the Gurage zone, Southern Ethiopia. According to the data obtained from the zonal administration, there are thirteen woredas and two town administrations in the zone. According to the 2017 Ethiopian central statistical agency population projection, the total population of the Gurage zone is 1,635,311; of 842,065 are females and the remaining 793,246 are males [22]. Historically, the Gurage people may be a complex mixture of Abyssinian, Harla, and other groups that migrated and settled in that region for different reason. The majority of the inhabitants of the Gurage Zone were reported as Muslims, with 51.02% of the population reporting that belief, while 41.91% practiced Ethiopian Orthodox Christianity, 5.79% were Protestants, and 1.12% Catholic [Gurage Zone Health Office, 2019/20].

The Gurage lives a sedentary life based on agriculture, involving a complex system of crop rotation and transplanting. Gurage people are known as hard workers and a model of good work culture in Ethiopia. Ensete is the main staple food, but other cash crops are grown, including coffee and khat, both traditional stimulants. The principal crop of the Gurage is "*Ensete*" (also Enset, "false banana plant"). This has a massive stem that grows underground and is involved in every aspect of Gurage life. It has a place in everyday interactions among community members as well as specific roles in rituals. In the Zone, the ethnicities of Amhara, Oromo, Wolayita, and Hadiya are found [Gurage Zone Health Office, 2019/20].

There are also seven hospitals (five public and two non-governmental) serving the total population in the zone. Five of the hospitals in the zone are primary hospitals and the remaining two are general zonal hospitals. All hospitals provided comprehensive emergency obstetric care services. Additionally, 72 health centers provide basic emergency obstetric care services in the Gurage zone. This study was conducted between April to May 2019/20.

## Study design and population

A community-based cross-sectional study design was conducted among women of reproductive age (15–49 years.) who had at least one history of childbirth and lived in the area for the last six months.

## Sample size determination

The sample size for the study was calculated using Epi Info™ version 7 StatCalc function of Sample Size calculation for population survey at 95% confidence interval (CI), 5% margin of error, considering 50.9% of mothers had harmful cultural practices during their pregnancy from a related study in Meshenti town, west Gojjam zone, North West Ethiopia [21] which provide the largest sample size and adding 10% non-response rate, a total of 422 study participants were estimated for this study.

## Sampling technique

From the woreda of the zone, five of them and one town administration were selected by a simple random sampling technique using the lottery method. Three kebeles from each woreda and two kebeles from Butajira town were randomly selected. Households with pregnant women were listed out from the family folder of health extension workers (HEW) and the study participants were selected using a simple random sampling technique with Excel generated random numbers. The total sample size was allocated proportionally to the selected kebeles and towns based on the number of pregnant women in their respective kebeles (See Fig 1).

## Data collection techniques and procedure

Data were collected by six experienced and trained data collectors who were Bachelor's degree holders using a structured questionnaire with face to face interviews. Training was provided for data collectors and supervisors regarding the objective of the study, data collection tool, ways of data collection, checking the completeness of data collection tools and how to maintain confidentiality. So then, they were acquiring a good knowledge about data collection. Moreover, the principal investigator, supervisors and the data collectors were a meet together for a discussion about the data collection process every day after data collection so that during this time we were discussed about the activity and the challenges and problems faced during their time of data collection every day, and we solved together what they faced. As a result we have maintained the consistency between different data collectors during the data collection period. After reviewing relevant works of literature from previous related studies and other materials, the questionnaire was prepared in English, translated to local language (Amharic), and administered with the Amharic version to facilitate understanding. One day training was provided for the data collectors and the supervisors, the questionnaire was pretested a week before the actual survey in a comparable setting in Silte town on 5% of the calculated sample size, after which the necessary correction and modification were made accordingly. The filled questionnaires were checked daily for completeness and internal consistency.

## Operational definitions/Term definition

**Perinatal period.**    Period including pregnancy, childbirth, and postnatal period.

**Harmful cultural practices.**    Having practiced any one of the following is considered as harmful cultural practices.

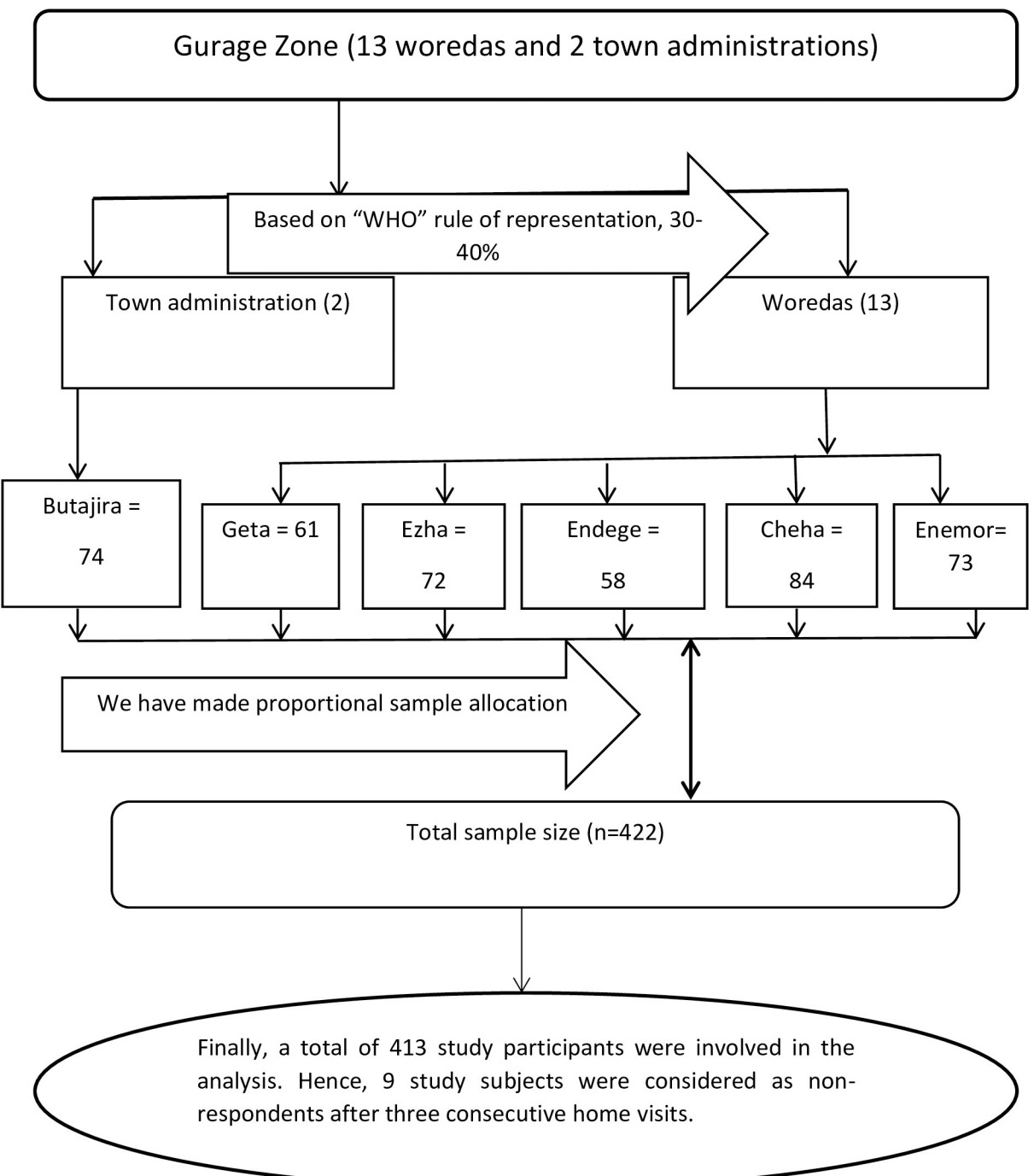

**Fig 1. Study flow chart/sampling procedure for a study on harmful cultural practices during pregnancy, childbirth, and postnatal period, and associated factors that clarify the sample enrolled or drop out at each phase in Southern Ethiopia, 2019.**

- Food taboos, Avoiding colostrum, Delay initiation of breastfeeding, giving butter and/or water for a newborn(pre-lacteal feeding), Use of "*kosso*" and Use of "*telba*"

- Abdominal massage

- Cutting umbilical cord by unsterile sharp materials, tie umbilical cord by unclean materials

    ■ Home delivery, early bath.

    **Food taboos** are a condition when women are abstaining from food and/or beverage consumption due to religious and cultural reasons during the perinatal period.

## Data processing and analysis

The collected data were entered into Epi Info version ™ 7 and exported to Statistical Package for Social Science (SPSS) version 25 for cleaning and analysis. Descriptive statistics were performed and the findings were presented with text and tables. Binary logistic regression was used to assess the association between each independent variable and outcome variable. Hosmer-Lemeshow statistic and Omnibus tests were done for model fitness. All variables with P < 0.25 in the bivariate analysis were included in the final model of multivariable analysis to control all possible confounders. Variables those were significant in previous studies and from a context point of view were included in the final model even if the above criteria were not meet. Normality test was done by using graphically and numerically methods. Based on the aforementioned methods, in the Q-Q Plot test of normality the data points were close to the diagonal line, the data was normally distributed. In histogram shape, approximates a bell-curve shape was observed that the data may have come for a normal population. Besides, in Shapiro-Wilk Test the value of the Shapiro-Wilk Test was greater than 0.05, the data is normal.

    Multicolinearity reduces the power of coefficients and weakens the statistical measure to trust the p-values to identify the significant independent variables. Hence, we would not be able to examine the individual explanation of the independent variables on the dependent variable. Therefore, in this study the multicolinearity problem was detected with the help of tolerance and its reciprocal, called variance inflation factor (VIF). Hence, variable with a value of tolerance is less than 0.1 and, simultaneously, the values of VIF 10 and above were checked accordingly. Unfortunately, no variable was detected that has collinearity effect or variable which has tolerance test less than <0.1 and VIF>10. The direction and strength of statistical association were measured by an odds ratio with 95% CI. Adjusted odds ratio along with 95% CI was estimated to identify factors for harmful cultural practices. In this study P-value < 0.05 was considered to declare a result as a statistically significant association.

## Ethical consideration

Ethical clearance was obtained from the Wolkite University College of Health and Medical Science Institutional Health Research Ethical Review Committee. An official letter was sent to the Gurage health office and the data collection was begun after permission and a cooperation letter was written to all districts on which the study was carried out. The study, purpose, procedure and duration, rights of the respondents and data safety issues, possible risks and benefits of the study were clearly explained to each participant using the local language. Then, all subjects were provided their informed written consent for inclusion before they participated in the study.

## Result

### Socio-demographic characteristics of study participants

A total of 413 study participants were involved in this study, making a response rate of 97.9%. The mean age of study participants was 27.6 (SD ± 5.4 years). More than half, 232(56.2%) of respondents were Orthodox followers by religion. More than four-fifth, 342 (82.8%) of the participant were Gurage by ethnicity. Almost two-third, 278(67.3%) of the participant were from rural areas. Almost three-fifth, 278 (60.5%) of the respondents had no formal education.

Nearly three-fourth, 299 (72.4%) of the study participants were self-employed by their occupation. Nearly two-third, 268(64.9%) of the respondents had ≥ 1000 Ethiopian birr average monthly income (See Table 1).

## Obstetrics characteristics of study participants

Nearly three-fifth, 227(55.0%) of the respondents were given 2–4 childbirth. Almost half, 216 (53.7%) of the participants were attended by a trained health professional during the last childbirth. Nearly three-fifth, 229(55.4%) of the participant had ANC follow-up during the last pregnancy. More than three-fourth, 187(81.7%) of the participants had taken antenatal care in the government health facility. Almost half, 213(51.6%) of the participants had taken less than 5km to reach the nearby health facility.

## Harmful cultural practices during the perinatal period

In this study, 295 (71.4%) of participants reported that they undertook some form of harmful cultural practices during the perinatal period. Regarding food taboos, 183 (44.3%) of the participants had consumed food taboos. Nearly one-fourth, 102 (24.7%) of the participants applied abdominal massage with butter to facilitate labor.

Nearly half, 193 (46.7%) of the participants were drunk "*Koso*" during pregnancy. Nearly half, 198 (47.9%) of the participants were drunk "*telba*" during pregnancy. Concerning birthplace, 197

**Table 1. Socio-demographic characteristics of study participants in Southern, Ethiopia, 2019 (n = 413).**

| Variable | Frequency | Percent |
|---|---|---|
| **Age (Year)** | | |
| 15–24 | 115 | 27.8 |
| 25–34 | 242 | 58.6 |
| ≥ 35 | 56 | 13.6 |
| **Ethnicity** | | |
| Gurage | 342 | 82.8 |
| Amhara | 22 | 11.9 |
| Oromo | 49 | 5.3 |
| **Marital status** | | |
| Married | 385 | 93.2 |
| Divorced | 17 | 4.1 |
| Windowed | 11 | 2.7 |
| **Residence** | | |
| Rural | 278 | 67.3 |
| Urban | 135 | 32.7 |
| **Education** | | |
| No formal education | 250 | 60.5 |
| Primary Education | 89 | 21.6 |
| Secondary and above | 74 | 17.9 |
| **Occupation** | | |
| Self-employed | 299 | 72.4 |
| Government Employed | 114 | 27.6 |
| **Average monthly income** | | |
| ≤500 | 41 | 9.9 |
| 501–999 | 104 | 25.2 |
| ≥1000 | 268 | 64.9 |

(47.7%) of participants were assisted by untrained TBA at home. Regarding umbilical cord care, 184 (44.6%) of respondents used an unclean blade to cut the umbilical cord and 174 (42.1%) of them used unclean thread to tie the umbilical cord. Regarding breastfeeding, 146 (35.4%) of the participants were provided pre-lacteal feeding (butter, honey, sugar, and water).

Nearly one-third, 131 (31.7%) of the participants discarded the colostrum (first yellowish milk). Nearly one-third, 120 (29.1%) of the participants were not provided breastfeeding within the first hour of birth. Regarding the initial time of bathing, 161 (39.0%) of the participants have not washed their babies within 24hr of birth (See Table 2).

## Factors associated with harmful cultural practices during the perinatal period

Multivariable analysis revealed that the odd of harmful cultural practices during the perinatal period were almost four [AOR 3.79, 95%CI, 1.97–7.28] times higher in women who had no

**Table 2. Harmful cultural practices during the perinatal period in Southern Ethiopia, 2019 (n = 413).**

| Variable | Frequency | Percent |
|---|---|---|
| **Food taboos** | | |
| No | 230 | 55.7 |
| Yes | 183 | 44.3 |
| **Abdominal massage with butter** | | |
| No | 311 | 75.3 |
| Yes | 102 | 24.7 |
| **"Koso" drinking** | | |
| No | 220 | 53.3 |
| Yes | 193 | 46.7 |
| **"Telba" drinking** | | |
| No | 215 | 52.1 |
| Yes | 198 | 47.9 |
| **Place of Birth** | | |
| Home | 197 | 47.7 |
| Health facility | 216 | 52.3 |
| **An instrument used to cut the umbilical cord** | | |
| Unclean blade | 184 | 55.4 |
| Clean blade | 229 | 44.6 |
| **The material used to tie the cord** | | |
| Unclean | 174 | 42.1 |
| Clean | 239 | 57.9 |
| **Pre lacteal feeding** | | |
| No | 146 | 35.4 |
| Yes | 267 | 64.6 |
| **Colostrum feeding** | | |
| No | 131 | 31.7 |
| Yes | 282 | 68.3 |
| **Time to start breastfeeding** | | |
| Not within 1hr | 120 | 29.1 |
| Within 1hr | 293 | 70.9 |
| **The initial time of bathing** | | |
| Not within 24hr | 161 | 39.0 |
| Within 24hr | 252 | 61.0 |

formal education than those who had secondary education and above. Participants who were rural by residence were nearly five [AOR 4.41, 95%CI, 2.63–7.39] times more likely to perform harmful cultural practices during the perinatal period than those who were urban by residence. Regarding ANC follow-up, participants who had not attended ANC follow up during the last pregnancy were almost three [AOR 2.62, 95%CI, 1.54–4.48] times more likely to execute harmful cultural practices during the perinatal period. Participants who were attended by an untrained attendant during the last childbirth were almost three [AOR 2.67, 95%CI, 1.58–4.51] times more likely to perform harmful cultural practices than those who were attended by a trained attendant **(See Table 3).**

**Table 3. Factors associated with harmful cultural practices during the perinatal period in Sothern Ethiopia, 2019 (N = 413).**

| Variables | Harmful cultural practices | | COR (95%) | AOR (95%) |
|---|---|---|---|---|
| | Yes (%) | No (%) | | |
| **Educational status** | | | | |
| No formal education | 200(67.8) | 50(42.4) | 3.05(1.75–5.31) | **3.79(1.97–7.28)**[*] |
| Primary education | 53(18.0) | 36(30.5) | 1.12(0.60–2.09) | 0.93(0.45–1.95) |
| Secondary education and above | 42(14.2) | 32(27.1) | 1.00 | 1.00 |
| **Residences** | | | | |
| Rural | 226(76.6) | 52(44.1) | 4.16(2.64–6.54) | **4.41(2.63–7.39)**[**] |
| Urban | 69(23.4%) | 66(55.9) | 1.00 | 1.00 |
| **Occupation** | | | | |
| Self-employed | 219(74.2) | 80(67.8) | 1.37(0.86–2.18) | 1.49(0.85–2.63) |
| Government employed | 76(25.8) | 38(32.2) | 1.00 | 1.00 |
| **Age** | | | | |
| 15–24 | 86(29.2) | 29(24.6) | 1.52(0.76–3.05) | 1.26(0.56–2.80) |
| 25–34 | 172(58.3) | 70(59.3) | 1.26(0.68–2.34) | 1.34(0.66–2.72) |
| ≥35 | 37(12.5 | 19(16.1%) | 1.00 | 1.00 |
| **Income** | | | | |
| ≤500 | 34(11.5) | 7(5.9) | 1.92(0.82–4.52) | 1.69(0.64–4.42) |
| 501–999 | 69(23.4) | 35(29.7) | 0.78(0.48–1.27) | 0.94(0.53–1.68) |
| ≥1000 | 192(65.1) | 76(64.4) | 1.00 | 1.00 |
| **ANC follow up** | | | | |
| No | 146(49.5) | 38(32.2) | 2.06(1.32–3.23) | **2.62(1.54–4.48)**[***] |
| Yes | 149(50.5) | 80(67.8) | 1.00 | 1.00 |
| **Birth attendant** | | | | |
| Untrained attendant | 161(54.6) | 36(30.5) | 2.74(1.74–4.31) | **2.67(1.58–4.51)**[****] |
| Trained attendant | 134(45.4) | 82(69.5) | 1.00 | 1.00 |
| **Number of live birth** | | | | |
| 2–4 | 157(53.2) | 70(59.3) | 0.78(0.51–1.20) | 0.69(0.41–1.14) |
| ≥5 | 138(46.8) | 48(40.7) | 1.00 | 1.00 |
| **Health facility accessibility** | | | | |
| >5 km | 150(50.8) | 50(42.4) | 1.41(0.92–2.16) | 1.42(0.87–2.34) |
| ≤ 5 km | 145(49.2) | 68(57.6) | 1.00 | 1.00 |

[*]Significant with P<0.001,

[**]Significant with P<0.001,

[***]Significant with P = 0.012 and

[****]Significant with P<0.002.

## Discussion

In this study, the level of harmful cultural practices during perinatal was found to be 71.4%. This finding suggests that health care providers should take into account the potential risk of harmful cultural practices while assessing clinical health assessment during antenatal care visits, childbirth ad post-natal visits of a woman. Besides, for healthcare planners this is vital. This knowledge can be used to build relevant programs, channeling scarce resources to teaching what is needed as opposed to imparting messages that are already known.

This level of harmful cultural practices is higher than what is reported from a study done in Meshenti town, west Gojjam zone, Amhara region, Northwest Ethiopia, and Cambodia [21, 23]. The discrepancy of these findings might be attributed to the difference in method used and study settings, sociodemographic characteristics of the study participants, and the availability and accessibility of the health services infrastructures. The report in this study implies that there is a lack of balance that the zone health office and regional health bureau could work in collaboration with the local health caregiver to lessen the harmful cultural practices of women during the perinatal period.

This finding was lower than the study conducted in Axum Town, North Ethiopia [20, 24]. The difference can be explained by the discrepancy in the background of the study participants and method used and study settings, time gap, and the availability and accessibility of the infrastructures. In addition to this, the health system-related factors might be contributing to this difference due to the extensive work of health extension workers and various health care institutions in awareness creation about the drawback of harmful cultural practices in the study area. The finding implies that there is crated platform for maternity care providers that could help them to be aware of local values, beliefs, and traditions to anticipate and meet the needs of women, gain their trust and work with them.

In this study, we have found several factors associated with harmful cultural practices during the perinatal period. These include having no formal education, being a rural residence, had no ANC follow-up for the last pregnancy, and attended by an untrained attendant.

Women who had no formal education were almost 4 times more likely to perform harmful cultural practices than women who had secondary education and above. Similar studies conducted in Nepal, Bangladesh, and Northwest Ethiopia have found that education level is an important factor of harmful cultural practice during the perinatal period [7, 8, 21]. This may be related to as women did not attend formal education they could not easily understand the drawback of harmful traditional practice on the health of the women themselves and their newborn baby. Moreover, women who had no formal education will not have better awareness about the benefits of preventive health care including avoiding harmful cultural practices and lower receptivity to new health-related information.

Regarding, the association of residency with harmful cultural practices during the perinatal period, women from rural settings were almost five times as likely to engage in harmful cultural practices in the perinatal period as those from urban settings. This is in-line with the fact that women that are rural residence will not have information that could assist them in making decisions regarding healthy behaviors including maternal and child health education and promotion. Hence, women who have rural residency will have a lack of access and availability of infrastructures like mass media and others that could enable them to be aware of the disbenefit of harmful cultural practices during the perinatal period. This is in line with a study conducted in Northern Ethiopia and North Karnataka [20, 25].

In this study, women who had no sought antenatal care during the last pregnancy were almost 3 times more likely to practice cultural misbehaviors than women who had sought antenatal care visits. The finding of this study was comparable with the findings of studies

conducted in Ethiopia, UK, and Taiwan [26–28]. This could be because women who had no previous ANC follow-up during the last pregnancy will not be aware of the drawback of harmful traditional practice on the health of women and the fetus during the perinatal period. In the Western Region of Ghana, traditional beliefs and practices, as well as negative attitudes of health workers, are found to reduce health utilization by pregnant women. Hence, health education concerning traditional practices that are detrimental to the health of pregnant women should be emphasized during ANC visits [29].

Concerning, the association of attended by an untrained attendant with harmful cultural practices during the perinatal period, those women who have attended by untrained attendant were almost 3 times more likely executing cultural misbehaviors than those who have attended by a trained attendant. This is in-line with the fact that women that are attended by untrained attendant will not have contact with the healthcare provider in the health facility during the perinatal period. Which in turn the women will not have information about the effect of harmful cultural practice on the health of the women themselves and their infants in the MCH clinic. The finding of this study is consistent with the studies done in Cambodia and Southwest Ethiopia [23, 30–31]. The strengths of this study include to the best of our knowledge, this is the first study carried out in the study area specifically in Gurage Zone, Ethiopia. The study has employed a nationally validated harmful cultural practices assessment tool.

The limitation of this study includes the study might be subjected to recall bias because the mothers failed to remember what they did during the perinatal period. This was minimized by probing the respondents about the event. Due to the cross-sectional nature of this study, establishing a true cause and effect relationship between harmful cultural practices and associated factors would be impossible.

## Conclusion

Harmful cultural practices were found to be high in the study area. In this study, having no formal education, being a rural residence, attending no ANC follow up for the last pregnancy, and attended by untrained attendant were factors significantly associated with harmful cultural practices during the perinatal period. Primary health care workers should take up the active role of women's health education on the importance of ANC visits. Besides, apart from primary health care workers, other stakeholders in the maternal health sector should also create awareness among women on those services in which they could actively get MCH clinic. The government should come up with policies that are helping to promote women's formal education and childbirth by a trained attendant.

## Supporting information

**S1 File. English and Amharic version questionnaire.**
(PDF)

**S2 File. Minimal data set.**
(SAV)

## Acknowledgments

We would like to acknowledge Wolkite University College of medicine and health science for approving the research project. Furthermore our special appreciation goes to data collectors for their genuine effort to bring reliable data. Finally we would like to whole heartedly acknowledge study participants without them this work could not be realized.

## Author Contributions

**Conceptualization:** Haimanot Abebe, Girma Alemayehu Beyene, Berhanu Semra Mulat.

**Data curation:** Haimanot Abebe, Girma Alemayehu Beyene.

**Formal analysis:** Haimanot Abebe, Berhanu Semra Mulat.

**Investigation:** Haimanot Abebe.

**Methodology:** Haimanot Abebe, Girma Alemayehu Beyene, Berhanu Semra Mulat.

**Project administration:** Haimanot Abebe.

**Resources:** Girma Alemayehu Beyene.

**Software:** Haimanot Abebe, Berhanu Semra Mulat.

**Supervision:** Haimanot Abebe, Berhanu Semra Mulat.

**Validation:** Haimanot Abebe.

**Visualization:** Haimanot Abebe, Girma Alemayehu Beyene.

**Writing – original draft:** Haimanot Abebe, Girma Alemayehu Beyene, Berhanu Semra Mulat.

**Writing – review & editing:** Haimanot Abebe, Girma Alemayehu Beyene.

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
