## [Decision Letter · Decision Letter 0]

22 Jan 2021

PONE-D-20-34446

Cultural malpractice during perinatal period and associated factors among women of child bearing age in Southern Ethiopia: Community based cross-sectional study

PLOS ONE

Dear Dr. Abebe,

Thank you for submitting your manuscript to PLOS ONE. After careful consideration, we feel that it has merit but does not fully meet PLOS ONE’s publication criteria as it currently stands. Therefore, we invite you to submit a revised version of the manuscript that addresses the points raised during the review process.

We look forward to receiving your revised manuscript.

Kind regards,

Nülüfer Erbil, Ph.D, Prof.

Academic Editor

PLOS ONE

Journal Requirements:

3.Thank you for stating the following financial disclosure:

4. We note that you use the term "cultural malpractice" throughout your manuscript and title. As the term "malpractice" is tied to specific legal meanings, we request that you change this term throughout the manuscript to something that does not contain the word "malpractice".

Reviewers' comments:

Reviewer's Responses to Questions

**Comments to the Author**

1. Is the manuscript technically sound, and do the data support the conclusions?

Reviewer #1: Yes

Reviewer #2: Yes

Reviewer #3: Yes

2. Has the statistical analysis been performed appropriately and rigorously? 

Reviewer #1: Yes

Reviewer #2: Yes

Reviewer #3: Yes

3. Have the authors made all data underlying the findings in their manuscript fully available?

Reviewer #1: Yes

Reviewer #2: Yes

Reviewer #3: Yes

4. Is the manuscript presented in an intelligible fashion and written in standard English?

Reviewer #1: Yes

Reviewer #2: Yes

Reviewer #3: No

5. Review Comments to the Author

Reviewer #1: Dear author,

The necessary corrections for the article are shown on the text. Please review the abstract section. There is an inconsistency in your sample size. I can only write in one language and admire anyone who attempts to write in a language other than their native one. unfortunately the English used here sometimes obscures your meaning. There is incorrect use of the definite and indefinite article, tenses, spelling and plural/singular to name a few problems. The manuscript should have been reviewed by a native English speaker

Reviewer #2: Review Comments for Author

Title: Cultural malpractice during perinatal period and associated factors among women of child bearing age in Southern Ethiopia: Community based cross-sectional study.

Investigating this research has many merits for multicultural and multi ethnic country Ethiopia and has a contribution for policy makers and other stakeholders who work up on community health.

Here below are the comments and questions for author;

Keywords should be no more than 5 and no less than 3.

“Perinatal period is considered as time period including the pregnancy, childbirth and postnatal period” what is implication of stating term definition in abstract?

Your method in abstract lacks overview of sampling technique, data presentation and analysis.

As far as this study has two objectives, it should have to consider sample size calculation for second objective too.

“Perinatal period: Time period including the pregnancy, childbirth and postnatal period” is term definition or operational definition? Clarify?

“The data collected were entered into Epi Info version ™ 7 and exported to Statistical Package for Social Science (SPSS) version 25 for cleaning and analysis” why? Cleaning and analysis was also possible for Epi info.

You stated multivariate and multivariable analysis interchangeably. Do you think it is similar? Which one you used? Justify?

“The mean age of study participants were 27.6 (SD ± 5.4 years)”. Did you check normality test? If yes how? Why you select mean to describe age? Why didn’t other descriptive statistics?

Are these ethnic groups only dwell in the study area?

“Koso”, “telba”… is local language and it should have to be elaborated and italic while you write.

In table 2, Frequency of food taboo is beyond total sample size? Do you have a justification? Why pre-lacteal feeding percentage left blank? Justify?

“Regarding ANC follow up, participants who had ANC follow up during the last pregnancy were almost three [AOR 2.62, 95%CI, 1.54-4.48] times more likely execute cultural malpractice during perinatal period.” How do you justify this? It contradicts the logic and the science? Is there possible explanation for your study?

Food taboo is a vague word. How you assessed it?

“This level of cultural malpractices is higher than what is reported from a study done elsewhere in Ethiopia” and “This finding was lower than the study conducted in Northwest Ethiopia” these two sentences are contradicting? Justify?

Your study was community based cross-sectional study and asks retrospectively about perinatal period. This is prone your study to recall bias. How did you reduce recall bias for this study?

Is your study had no limitation? if yes add limitations of the study

Reference no.14 and no.18 are similar. Consider revision

Reviewer #3: 1-It was good to see researchers are trying to address an important issue of developing country

2- It would be appreciated, if author can explain the reasons of repeating the similar study in different zone of Ethiopia, when it was already conducted in north west Ethiopia (mentioned in sample size reference)

3- It would be nice, if STROBE guidelines can be followed so that methodology can be fully evaluated.

4- The whole document needs a substantial language revision to improve grammar and style.

5- The whole document should be revised considering the within text citation as it has been noticed at multiple places where references within text references are missing like page 3, paragraph 1, line 2 and 6 and paragraph 5 line 2, 4 etc.

6- Study setting needs further description in terms of population being catered like their ethnicity, socio economic status so one can understand the homogeneity or heterogeneity among the settings

7- Kindly justify why are you calling the the sampling technique as simple random sampling and not proportionate or multi level sampling

8 - How did you maintain the consistency between different data collectors? as there were 6 separate data collectors

9- who did training for data collectors and what was covered in training?

10- Authors have mentioned that corrections and modification was made after pretesting, please specify

11- Why the results considered confounding variable? as this study does not have any main exposure so please share the reasons of considering confounders as it is not meeting the criteria of assessing confounders.

12- did the author assessed the normality of each variable?

13- data results were based on 413 whereas in abstract it was mentioned 422. please correct this and explain the reasons of dropout?

14- did researcher did the stratified analysis on different perinatal time i.e pregnancy, childbirth & post natal? this would be a great addition to see at which phase culture malpractices affects the most.

15- On Page 9 in table 2, percentages are missing for pre lacteal feeding.

16- Does researcher assess multi collinearity between independent variables?

17- Can these finding be generalized, justify and write yes or no with proper justification

18- Author should highlight the strengths and limitation of this study? Any bias ( selection or reporting) observed & how it was managed?

6. PLOS authors have the option to publish the peer review history of their article (what does this mean?). If published, this will include your full peer review and any attached files.

Reviewer #1: No

Reviewer #2: **Yes: **Alex Yeshaneh

Reviewer #3: **Yes: **Shireen Shehzad Bhamani

---

## [Author Response · Author response to Decision Letter 0]

9 Mar 2021

First of all, the authors would like to thank “PLOS ONE” Journal editors and the respective reviewers for reviewing our manuscript and providing the necessary comments to be corrected and more scientifically acceptable. Again the authors would like to thank all reviewers who are involved in reviewing our manuscript with a great dedication and responsibility. We understand your help and support to make our manuscript more scientifically sound and all the comments raised by the editor and the reviewers were incorporated in the revised submission. 

The authors declare there is no external funding source obtained for this study and statement which reads as "a funders had no role in study design, data collection and analysis, decision to publish, or preparation of the manuscript" is removed from the cover letter and the revised version of the manuscript. Thank very much for your time and consideration!!

---

## [Decision Letter · Decision Letter 1]

11 May 2021

PONE-D-20-34446R1

Cultural harmful practices during perinatal period and associated factors among women of childbearing age in Southern Ethiopia: Community based cross-sectional study.

PLOS ONE

Dear Dr. Abebe,

Thank you for submitting your manuscript to PLOS ONE. After careful consideration, we feel that it has merit but does not fully meet PLOS ONE’s publication criteria as it currently stands. Therefore, we invite you to submit a revised version of the manuscript that addresses the points raised during the review process.

We look forward to receiving your revised manuscript.

Kind regards,

Nülüfer Erbil, Ph.D, Prof.

Academic Editor

PLOS ONE

Journal Requirements:

Reviewers' comments:

Reviewer's Responses to Questions

**Comments to the Author**

1. If the authors have adequately addressed your comments raised in a previous round of review and you feel that this manuscript is now acceptable for publication, you may indicate that here to bypass the “Comments to the Author” section, enter your conflict of interest statement in the “Confidential to Editor” section, and submit your "Accept" recommendation.

Reviewer #2: All comments have been addressed

Reviewer #3: All comments have been addressed

2. Is the manuscript technically sound, and do the data support the conclusions?

Reviewer #2: Yes

Reviewer #3: Yes

3. Has the statistical analysis been performed appropriately and rigorously? 

Reviewer #2: Yes

Reviewer #3: Yes

4. Have the authors made all data underlying the findings in their manuscript fully available?

Reviewer #2: Yes

Reviewer #3: Yes

5. Is the manuscript presented in an intelligible fashion and written in standard English?

Reviewer #2: Yes

Reviewer #3: No

6. Review Comments to the Author

Reviewer #2: All my questions and concerns are responded to well, conclusions are supported by data and now the manuscript scientifically sounds. Besides, the amendment of the title instead of cultural malpractice to harmful traditional practice also sounds more.

Reviewer #3: 1-I strongly feel that, study flow chart will be a great addition to this manuscript and it will clarify the sample enrolled or drop out at each phase.

2- Some of the comments were appropriately answered by author but I could not see these were incorporated in manuscript like normality assessment, training details and steps taken by researchers to improve consistency between data collectors. Plus considering multicollinearity ( do share which variables were found correlated) etc.

These points are their efforts to improve the rigor of the study so it should be added.

7. PLOS authors have the option to publish the peer review history of their article (what does this mean?). If published, this will include your full peer review and any attached files.

Reviewer #2: **Yes: **Alex Yeshaneh

Reviewer #3: **Yes: **Shireen Shehzad Bhamani

---

## [Author Response · Author response to Decision Letter 1]

13 May 2021

Editors: - Please review your reference list to ensure that it is complete and correct. If you have cited papers that have been retracted, please include the rationale for doing so in the manuscript text, or remove these references and replace them with relevant current references. Any changes to the reference list should be mentioned in the rebuttal letter that accompanies your revised manuscript. If you need to cite a retracted article, indicate the article’s retracted status in the References list and also include a citation and full reference for the retraction notice. 

Authors Response: Dear Nullifier Erbil [(PHD, Prof.) Academic Editor of PLOS ONE], Thank you very much for your formative and productive comments. We have taken correction in the revised manuscript accordingly. Outdated references are omitted from the references lists and replaced by recently published articles or literatures. Accordingly the following outdated references are changed. 

1, United Nations (UN), Fact Sheet No.23, Harmful Traditional Practices Affecting the Health of Women and Children, UN Office of the High Commissioner for Human Rights (UNCHR), Editor. 2006: Geneva, Switzerland. Is changed by Foad HS, Katz R, Migration IO for. World Migration Report 2020 (full report) [Internet]. Vol. 54, European Journal of Political Research Political Data Yearbook. 2015. 1–18 p.

2. Alene, G.D. and M. Edris, Knowledge, Attitudes and Practices involved in Harmful Health Behavior in Demba District, northwest Ethiopia. Ethiop.J.Health. Dev, 2002. 16(2): p. 199-207. Is changed by Hadush Z, Birhanu Z, Chaka M, Gebreyesus H. Foods tabooed for pregnant women in Abala district of Afar region, Ethiopia: An inductive qualitative study. BMC Nutr. 2017;3(1):1–9.

3, Keno D: cultural practices during pregnancy & childbirth among WCBA in shebe town. In.: research report summated to the department of health officer as practical …; 1998. Is changed by Nana A, Zema T. Dietary practices and associated factors during pregnancy in northwestern Ethiopia. BMC Pregnancy Childbirth. 2018;18(1):1–8.

XXXX- Concerning English language and grammatical editing suggested by the reviewers. We have taken corrections concerning to any English errors and typos by the help of a man who have PHD in English language. Furthermore, we have also taken correction for any English errors using online grammar and English typo correctors apps (We used the following links- 

https://app.grammarly.com/?network=g&utm_source=google&matchtype=e&gclid=Cj0KCQjwo-aCBhC-ARIsAAkNQiuJ49UHhl6ibhQfzq9D4wGrbSOeZPv49UoRqSnd4ThQ-KKrPp uBp4aAjLgEALw_wcB&placement=&q=brand&utm_content=486649398671&gclsrc=aw.ds&utm_campaign=brand_f1&utm_medium=cpc&utm_term=grammarly 

 and

https://pubsure.researcher.life/author/?active_tab=recent_plan

So, now we have solved all iniquities related with the English language.......including the tense used and unnecessary capitalization and other typos/ errors 

Point by point response to (Reviewer # 3)

Thank you very much, Dear Sir/Madam (Reviewer # 3), we would like to give you our appreciation or thankfulness for your endless support to make our manuscript scientifically well-conditioned. Besides, we would like to thank you for sharing your knowledge and experiences in the whole reviewing of this manuscript. 

Reviewer # 3: I strongly feel that, study flow chart will be a great addition to this manuscript and it will clarify the sample enrolled or drop out at each phase.

Authors Response: Thank you very much, Dear Sir/Madam (Reviewer # 1), we have taken correction accordingly in the revised manuscript. 

Reviewer # 3: Some of the comments were appropriately answered by author but I could not see these were incorporated in manuscript like normality assessment, training details and steps taken by researchers to improve consistency between data collectors. Plus considering multicolinearity ( do share which variables were found correlated) etc.

Authors Response: Thank you very much, Dear Sir/Madam (Reviewer # 3), we have taken correction and incorporated all suggestion given accordingly in the revised manuscript.

---

## [Decision Letter · Decision Letter 2]

21 Jun 2021

Harmful cultural practices during perinatal period and associated factors among women of childbearing age in Southern Ethiopia: Community based cross-sectional study.

PONE-D-20-34446R2

Dear Dr. Abebe,

We’re pleased to inform you that your manuscript has been judged scientifically suitable for publication and will be formally accepted for publication once it meets all outstanding technical requirements.

Kind regards,

Nülüfer Erbil, Ph.D, Prof.

Academic Editor

PLOS ONE

Additional Editor Comments (optional):

Reviewers' comments:

Reviewer's Responses to Questions

**Comments to the Author**

1. If the authors have adequately addressed your comments raised in a previous round of review and you feel that this manuscript is now acceptable for publication, you may indicate that here to bypass the “Comments to the Author” section, enter your conflict of interest statement in the “Confidential to Editor” section, and submit your "Accept" recommendation.

Reviewer #3: All comments have been addressed

2. Is the manuscript technically sound, and do the data support the conclusions?

Reviewer #3: Yes

3. Has the statistical analysis been performed appropriately and rigorously? 

Reviewer #3: Yes

4. Have the authors made all data underlying the findings in their manuscript fully available?

Reviewer #3: Yes

5. Is the manuscript presented in an intelligible fashion and written in standard English?

Reviewer #3: Yes

6. Review Comments to the Author

Reviewer #3: Authors have responded well to every query raised at second review. Technically it is now a well written piece.

7. PLOS authors have the option to publish the peer review history of their article (what does this mean?). If published, this will include your full peer review and any attached files.

Reviewer #3: **Yes: **Shireen Shehzad Bhamani

---

## [Editor Report · Acceptance letter]

23 Jun 2021

PONE-D-20-34446R2 

Harmful cultural practices during perinatal period and associated factors among women of childbearing age in Southern Ethiopia: Community based cross-sectional study. 

Dear Dr. Abebe:

I'm pleased to inform you that your manuscript has been deemed suitable for publication in PLOS ONE. Congratulations! Your manuscript is now with our production department. 

Kind regards, 

on behalf of

Dr. Nülüfer Erbil 

Academic Editor

PLOS ONE